# Digital twin-enabled supply chain management with visibility and traceability: a case study*

Yishu Yang, Ming Li, Hang Wu, Yaqi Dai, Ray Y Zhong* *Member, IEEE*

*Abstract*—**Effective management of the building materials supply chain is of utmost importance in construction projects, particularly in cities like Hong Kong characterized by high-rises, high population density, and hilly terrains. Given these factors, visualizing the construction materials supply chain becomes imperative. However, existing research primarily focuses on monitoring logistics road conditions and logistics network, neglecting the need for detailed logistics data on individual objects. This limitation hampers the overall performance improvement of the supply chain. To address this gap, this paper proposes a digital twin-supported supply chain management framework that enhances visibility and traceability. The framework integrates advanced technologies such as digital twins, Building Information Model, Global Positioning System (GPS), and smart sensors, enabling visual project management throughout the construction process. This includes visualization of the supply chain and logistics, as well as material visibility and traceability. To validate the feasibility of the framework, a case study was conducted using a water pump house project in Hong Kong. The framework allows users to monitor project progress by tracking material procurement and delivery, monitoring the local distribution of materials on construction sites, and scheduling construction materials. Ultimately, the proposed framework has the potential to benefit the general public in Hong Kong as end-users of construction projects.**

## I. INTRODUCTION

The transformation of dynamic supply chains within the prefabricated construction sector presents new challenges and opportunities. Dynamic supply chains necessitate higher standards for data quality, reliability, and timely acquisition to facilitate real-time, data-driven decision-making [1]. Ensuring information traceability, reducing information fragmentation, enhancing interoperability, and improving reliability are inherent challenges in managing dynamic supply chains for prefabricated buildings. The emergence of digital twins and the rapid advancement of smart technologies have garnered interest in their application within the construction industry [2]. A "digital twin" refers to a highly accurate digital representation of a physical asset, enabling two-way communication between the physical and digital entities [3]. It can monitor, simulate, predict performance, and control physical systems through bidirectional data-to-information flow throughout their lifecycle [4].

Digital twins have been widely used in the manufacturing industry. In the product manufacturing stage, Tao et al. [5]

Yishu Yang is pursuing a Ph.D. degree with the Department of Industrial and Manufacturing Systems Engineering, The University of Hong Kong, Hong Kong (e-mail: yangys@connect.hku.hk) .

Ray Y Zhong Department of Industrial and Manufacturing Systems Engineering, The University of Hong Kong, Hong Kong (e-mail: zhongzry@hku.hk).

proposed an implementation plan for a digital duplex workshop, outlining its system composition, operating mechanism, characteristics, and key technologies. This provides a theoretical foundation for achieving cyber-physical integration in manufacturing workshops. Zhang, et al. [6] introduced digital twin technology to enhance dynamic scheduling and explored methods for machine availability prediction, interference detection, and performance evaluation based on digital twins. Kong, et al. [7] proposed a data construction method applied to the workshop's digital binary system. These studies highlight the application potential of digital twins, which are increasingly being utilized in the construction industry. Manfren, et al. [8] proposed a lean and interpretable "digital twin" for monitoring building energy performance and demonstrated the scalability of the proposed energy signature model through a practical case study. Kang, et al. [9] developed a conceptual framework for smart building demolition that collects, maintains, and analyzes comprehensive information using smart BIM to maximize the recycling and reuse of demolition waste, allowing simulation of the demolition process through DT-supported BIM. Pal, et al. [3] proposed a framework for vision-based closed-loop construction control through digital twin construction. Sun and Liu [10] proposed a hybrid DT-BIM model to assist decision-making, progress updating, resource scheduling, and identification of resource shortages.

While these studies have proven the feasibility of digital twin applications in the construction field, there is a dearth of research on their visualization and traceability within the construction supply chain. Dyck, et al. [11] suggests that digital twins can enhance market traceability in post-harvest grain handling. Zhuang, et al. [12] proposed a digital twin-based method for managing complex product assembly data and process traceability, but lacked synchronous modeling based on the product assembly process to assist decision-making. Liu, et al. [13] proposed a digital twin-driven machining quality tracking and dynamic control method. These studies underscore the advantages of digital twins in achieving traceability. In the realization of dynamic physical environments, prefabricated building projects generate substantial non-geometric data. Although these data are crucial for decision-making, they are not fully utilized [14]. Brandín and Abrishami [1] proposed a conceptual framework that integrates Internet of Things (IoT), Building Information Modeling (BIM), and Blockchain (BCT) technologies to track the lifecycle of building asset data. The integration of these technologies provides a decentralized solution to enhance supply chain security and reliability. However, this research has not been implemented in real-world cases, and achieving standardization in off-site manufacturing requires extensive collaboration among stakeholders. Additionally, continuous

changes in BIM models may result in data redundancy, leading to a voluminous amount of data [15].

Considering the critical role of supply chain management in enabling the efficient flow of construction materials and services, a robust and reliable system platform that minimizes human error is necessary. This paper proposes a digital twin-supported supply chain management framework with visibility and traceability. The framework integrates advanced technologies such as digital twins, BIM, GPS, and smart sensors to achieve visual project management in construction, including supply chain and logistics visualization, and material visibility and traceability.

## II. MOTIVATING SCENARIO

There are three main phases in prefabricated construction, including scheduling, logistics, and on-site installation. Throughout the project, the main contractor coordinates and supervises the quality, cost, and schedule, encompassing the entire life cycle management of materials. Fig. 1 illustrates the three primary motivating scenarios in prefabricated construction: 1) Tracking of procurement deliveries, 2) Material local distribution tracking in construction sites, 3) Building material scheduling. These scenarios involve three main stages: material procurement, material location distribution, and project management.

### A. Scenario 1: Tracking of material procurement deliveries

In this scenario, construction materials are purchased by customers through an online construction material purchasing platform and subsequently delivered to the construction sites. The materials are sourced from various suppliers located in different cities and delivered by different third-party logistics companies, resulting in a complex logistics network. Without track and trace services, it becomes challenging to optimize the construction assembly plan effectively.

### B. Scenario 2: Material local distribution tracking in construction sites

In this scenario, construction materials are distributed to each floor or area as required. The ownership of materials is transferred during this distribution process, which involves multiple operators from different divisions. The frequent transfer of ownership increases the risk of incorrect deliveries due to the negligence of operators in their distribution tasks, and at times, materials become difficult to retrieve.

### C. Scenario 3: Building material scheduling

Scenario 3 focuses on the synchronization between building material scheduling. Traditionally, BIM aids designers and customers in building design but remains isolated from building material scheduling. While this approach allows for the development of design schemes, it fails to optimize building material scheduling to streamline construction projects. In most cases, construction projects are hindered by delays in material supply.

## III. A DIGITAL TWIN-ENABLED PLATFORM

### A. Framework

In the context of prefabricated construction, we proposed a service-oriented 5D digital twin model that offers decision-making support across various stages of the materials' supply chain, enabling visual real-time monitoring. The framework is shown in Fig. 2. This enriched model incorporates two additional dimensions: digital twin data and services. The proposed five-dimensional model comprises Digital Twin Data, Virtual Model, Events, Interactions, and Services, all interconnected through Events. The model can be expressed as: $M_{DT} = (DD, ES, VM, Ss, IR)$.

Where DD is fit-out construction DT data, ES is a collection of events, connecting the entire 5D model, VM are virtual fit-out buildings, Ss are the prefabricated construction services, and IR are the interactions. The specific content is introduced in the following section.

### B. Smart construction objects

The smart construction objects (SCO) located in the physical infrastructure layer where serves as the fundamental component of the entire platform, establishing an intelligent construction environment within typical prefabricated sites like warehouses, logistics centers, and construction sites. conventional construction resources, such as labor, machinery, and materials, are transformed into smart objects by integrating them with intelligent Internet of Things (IoT) devices. Unlike traditional construction objects, these smart construction objects (SCOs) possess three key attributes: awareness, communicativeness, and autonomy. Awareness refers to the SCO's capability to sense and record real-time conditions and the surrounding environment, which can be achieved through various sensing or positioning technologies. Communicativeness represents the object's ability to transmit information. For example, SCOs can proactively send event alerts on a regular basis. Communication technologies such as Wi-Fi, Bluetooth, ZigBee, and others facilitate this capability. Autonomy means that the SCO is able to act autonomously according to some preset rules. The implementation of this ability depends on the reasoning algorithm. Autonomy signifies the SCO's ability to act independently based on preset rules, and its implementation relies on reasoning algorithms. Generally, the relevant fit-out construction data is collected at this layer and can be retrieved and disseminated through a smart gateway to facilitate interactions with upper layers.

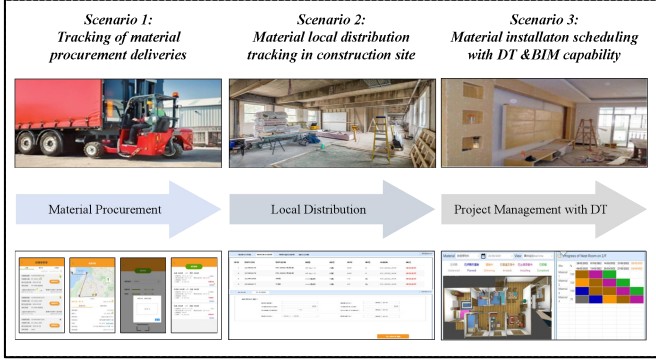

Figure 1. Primary motivating scenarios

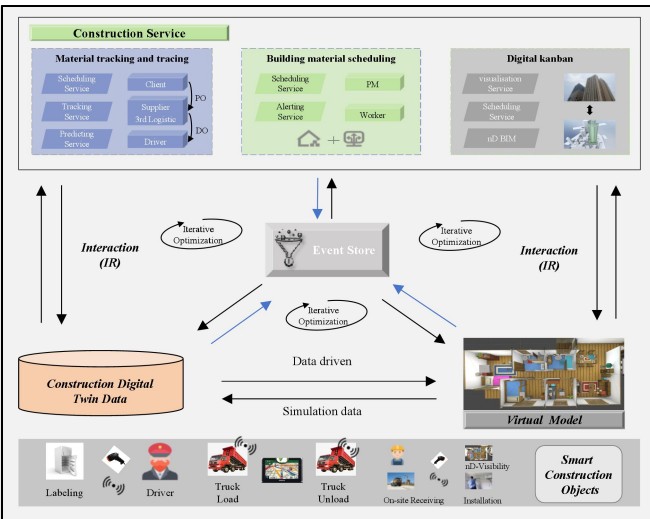

Figure 2. Framework of the DT-enabled platform

## C. Construction digital twin data

Data plays a pivotal role in driving the Digital Twin framework. The 5D DT model need to efficiently handle data that is characterized by multiple time scales, dimensions, sources, and heterogeneity. DD mainly contains four types of data, and can be described as . Dg is various SCO-related data collected from the Smart Gateway, including static attribute data and dynamic condition data. These data can be collected by multiple IoT sensors, embedded systems, data acquisition cards, and other means. Dg primarily reflects the physical attributes of the prefabricated construction project's entities, such as specifications, functions, and performance. It also captures real-time status information and dynamic processes like distribution and installation. Dv comprises data associated with the virtual building, generated by the virtual model. It encompasses geometric model-related data, including the dimensions of the building, assembly relationships, and positions within the entire fit-out project. Additionally, Dv includes data related to physical entity attributes, such as material properties and features, as well as simulation data for processes, behavior, evaluation, analysis, and prediction. Ds refers to data related to system services, specifically describing the invocation and execution of services. It encompasses algorithms, models, and other relevant data used in service provision and management. Di is derived data from the aforementioned data. The original data undergoes further transformations, preprocessing, or classification. For example, physical data may be integrated with multi-temporal data or historical data, resulting in more comprehensive information and enhanced information value.

## D. Construction event

Events serve as the primary medium in the construction 5D DT model, representing actions or changes in state within a monitored environment during a specified time frame. In the smart construction environment, when a construction element becomes SCO, its related information can be perceived. The information generated from the interaction between SCOs undergoes event processing, transforming it into basic events. By defining specific rules, these basic events can be combined to form complex events.

The proposed framework focuses on the entire construction material supply chain process, starting from the main contractor placing an order until the project is completed. This process can be considered as an event-driven business process and can be effectively modeled using the Event-driven Process Chain (EPC) methodology. The EPC includes process start events, process middle events, and process end events. A change in state triggers an event, which, in turn, drives specific behavior based on predefined rules, subsequently leading to the occurrence of the next event. Fig. 3 presents a partial EPC example illustrating the process from the submission of an order for fit-out materials to its completion.

## E. Virtual construction model

Similar to the virtual entity in the traditional data twin model, virtual fit-out construction represents physical data in a virtual space. The geometry, physics, behavior and rule models of virtual space are described using three-dimensional models and associated events. These four types of models are assembled, integrated, and fused to achieve a multi-temporal and multi-spatial characterization and description of physical layers, encompassing events and entities. The virtual construction can accurately replicate various construction progress at multiple hierarchical levels within the construction site. It enables twin-level simulation and monitoring of construction activities, aligning with the construction logic and meeting the site's requirements. Consequently, it facilitates the visualization of physical data related to construction activities, supporting further analysis, optimization, and decision-making.

## F. Construction service

The construction service is constructed upon the data layer and the virtual layer, encompassing the service-based encapsulation of data, models, algorithms, and simulations within the digital twin application process. This service system caters to the diverse needs of users across various fields, providing business services in the form of software and mobile applications (APPs). Within the proposed 5D digital twin model, the service system comprises three modules: the material tracking and tracing system, the building material scheduling system, and the digital kanban system. Leveraging data, model-related technologies, and optimized algorithms, the construction service system facilitates multidimensional real-time monitoring, as well as multi-angle simulation analy-

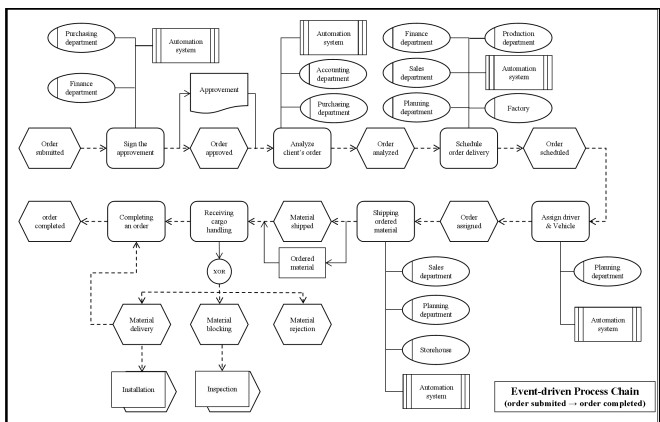

Figure 3. A partial EPC example

-sis of the entire material process, based on the specific requirements and actual situation during construction. The service system offers role-specific visualization and status updates at different stages through various means, effectively filtering information to ensure its validity. By shielding the internal heterogeneity and complexity of the digital twin, the service system delivers standardized outputs to users, enabling convenient on-demand utilization.

*G. Interaction*

Interaction refers to the action that occurs between two or more parties, wherein they can influence one another. These interacting parties are interconnected and interdependent. In the context of the 5D digital twin model, interaction facilitates the interconnection and intercommunication of each component. Events generate a vast amount of data, which is transmitted to the Data Distribution (DD) module through protocol specifications. The processed data or instructions within the DD module are then fed back to the event set, driving the occurrence of new basic or complex events. Similarly, events drive the generation of data, which is transmitted to the Simulation system (Ss) module in real-time, enabling Ss updates and optimization. The analysis, decision optimization, and other results produced by the Ss module are provided to specific users in the form of software or mobile applications (APPs), facilitating construction process control through manual operation of the event set. Two-way communication between the Virtual Model (VM) and the Ss module is achieved through a software interface, encompassing instruction transmission, message synchronization, and other functionalities. The DD module transmits fusion data, associated data, life cycle data, etc., to the VM module through the database interface, driving dynamic simulation. The simulation analysis data generated by the VM module are then fed back to the DD module to support data processing and generate optimization instructions. The interaction between the Ss module and the DD module follows a similar pattern. Real-time storage of Ss data to the DD module is facilitated through the database interface, while the Ss module simultaneously accesses real-time data and algorithms from the DD module to support its own operations and optimization.

## IV. CASE STUDY

A case study was conducted in collaboration with Chun Wo Development Holdings Limited. The DT-enabled platform is validated based on the Jockey Club pump house project located in Hong Kong, where RFID tags are deployed on the construction materials, and a large number of IoT information collection devices are deployed to collect dynamic information on materials in real-time. Considering the transportation environment and economic factors of the materials, RFID tags and readers were chosen as IoT devices. Contractor used existing software for procurement to obtain time-series data related to purchase orders. RFID tags were deployed on materials before their outbound transportation. The qIDmini R1279I wearable device was selected for reading the RFID labels. Real-time event information was captured and organized into an event log. The event log is the properly formatted time-series data and used to track detailed construction flow.

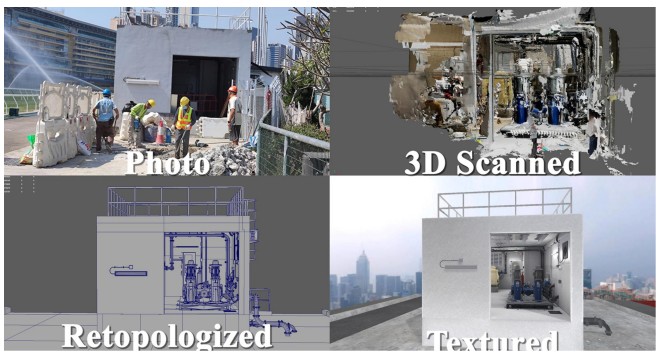

Figure 4. Initial virtual model development

The system effectively manages real-time dynamic data and historical data of materials, enhancing the traceability and visualization of building materials. Initially, a basic virtual model is created using Unity 3D, as illustrated in Fig. 4. Subsequently, web-based high-fidelity digital twins are generated from the Unity 3D engine using WebGL tools, enabling real-time status updates of materials. The different statuses include unplanned, planned, delivering, in-stock, installing, and completed, as depicted in Fig. 5. The real-time status of materials within the virtual model is represented by distinct colors, providing users with a comprehensive overview of the materials' progress throughout the project implementation. Active devices are registered in the Devices Registration Directory. System changes, such as the creation of work activities, task priority modifications, and task execution, trigger system messages that are sent to the notification service via the message gateway, notifying different stakeholders. These messages are then transmitted to the target devices through various push networks based on the target platform.

Additionally, material-related traceability events are displayed in the virtual model. The basic material information, such as material id, related PO id, and DO id, is linked together. The system page presents the timestamps of significant events specific to each material, including distribution start time, planned warehousing time, and actual warehousing time, as shown in Fig.5. This enables project managers to gain a com-

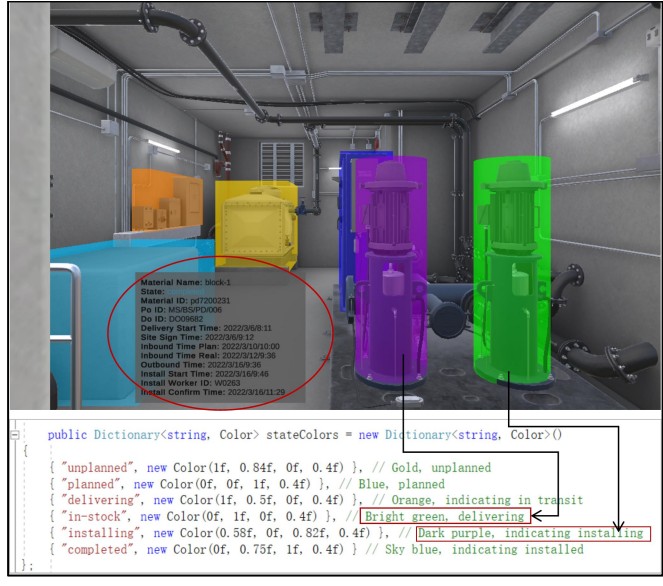

Figure 5.Material visualization

-prehensive understanding of the entire supply chain process for specific materials and make informed rescheduling decisions in a timely manner. The implementation of material information tracking and real-time status updates is demonstrated in Fig. 6.

## V. PRACTICAL SIGNIFICANCE AND CONTRIBUTIONS

The proposed platform establishes an IoT environment to provide specific data inputs of relevant resources throughout the entire process, rather than burdening a single stakeholder (typically the general contractor) with the heavy load of data input. This platform combines the benefits of traditional data sources and DT data to support material construction management while causing minimal disruption to existing workflows. Through the proposed platform, all stakeholders of the projects can be interconnected to support collaboration.

Different stakeholders can benefit from collaborative efforts. Logistics companies improve their services through dynamic vehicle tracking and visualized material deliveries, resulting in more efficient logistics and supply chain management. Major contractors benefit from real-time information on materials, enabling them to be more flexible when facing project changes. In addition, there is rich and real time record of materials delivered to sub-contractor by main contractor. These drastically reduces paperwork and disputes about materials handover. The proposed platform also tracks material being allocated to the right spot for actual work being carried out, e.g., certain pricey fixtures (cabinet, basin, door frame, etc.) can now be seen arriving at the work site down to

```csharp
public void UpdateObjectData(string json)
{
    ObjectDataUpdate updateData = JsonUtility.FromJson<ObjectDataUpdate>(json);
    if (dataDictionary.ContainsKey(updateData.MaterialName)) // Using MaterialName as the key
    {
        var objectData = dataDictionary[updateData.MaterialName];
        // Update all properties
        objectData.MaterialID = updateData.MaterialID;
        objectData.PoID = updateData.PoID;
        objectData.DoID = updateData.DoID;
        objectData.DeliveryStartTime = updateData.DeliveryStartTime;
        objectData.SiteSignTime = updateData.SiteSignTime;
        objectData.InboundTimePlan = updateData.InboundTimePlan;
        objectData.InboundTimeReal = updateData.InboundTimeReal;
        objectData.OutboundTime = updateData.OutboundTime;
        objectData.InstallStartTime = updateData.InstallStartTime;
        objectData.InstallWorkerID = updateData.InstallWorkerID;
        objectData.InstallConfirmTime = updateData.InstallConfirmTime;
        objectData.State = updateData.State;

        // Update color based on the new state
        ApplyColorBasedOnState(updateData.MaterialName, updateData.State);
    }
}

private void ApplyColorBasedOnState(string objectName, string state)
{
    if (stateColors.TryGetValue(state, out var color))
    {
        var objectInScene = GameObject.Find(objectName);
        if (objectInScene)
        {
            var renderer = objectInScene.GetComponent<Renderer>();
            if (renderer)
            {
                renderer.material.color = color; // Applying the new color
            }
        }
    }
}

private string ColorToHex(Color color)
{
    int r = Mathf.RoundToInt(color.r * 255);
    int g = Mathf.RoundToInt(color.g * 255);
    int b = Mathf.RoundToInt(color.b * 255);
    int a = Mathf.RoundToInt(color.a * 255); // Including Alpha channel
    return $"#{r:X2}{g:X2}{b:X2}{a:X2}"; // Convert to hexadecimal format
}
```

Figure 6. Implementation of material information tracking

floor, unit and even room level. This also helps indicate an important piece of information both contractor and subcontractor need – Progress Data. Both parties can use it to estimate how works are being progressed. Other stakeholders also benefit from real-time information access. The proposed system provides when, where and items' identification to the buyer automatically and digitally, fulfil exactly what the buyer needs and wants, thus enhance the suppliers' customer satisfaction, a key to enhance their competitiveness and business. Visibility and traceability tools enable real-time monitoring of material status. The hierarchical and multidimensional information provided by the platform contributes to the management of the entire supply chain. Additionally, the data stored in the platform continuously influences the training of neural networks within the system, leading to more accurate models and optimized system performance.

## VI. CONCLUSION

Prefabricated construction requires more detailed and multidimensional data to effectively monitor the entire process and make informed decisions. Real-time monitoring enables project managers to optimize and adjust building materials logistics schedules on the fly. Visualization of construction material logistics has emerged as a research area that supports planning and scheduling decisions in construction projects. However, existing construction visualization services primarily collect logistics data at key ownership transfer points, which are established between different parties. As a result, logistics activities and the status of building materials within the same party cannot be effectively monitored and visualized due to insufficient, inadequate, and incomplete data. Consequently, this limitation poses significant challenges in tracking, tracing, and dispatching building materials. To address this issue, this paper presents a digital twin-supported supply chain management framework with enhanced visibility and traceability. The framework integrates advanced technologies, such as digital twins, BIM, GPS, and smart sensors, to enable visual project management throughout the construction process, including supply chain and logistics visualization, as well as material visibility and traceability. The proposed platform deploys IoT devices to collect real-time data on the actual status of relevant operations, forming a closed-loop visibility and traceability mode, allowing different end users to monitor project status in real time. Different parties can focus on various aspects throughout the lifecycle of the prefabricated fit-out construction project and can collaboratively address issues or handle unexpected deviations using real-time visibility tools.

The feasibility of the framework was validated through a case study conducted on a water pump house project in Hong Kong. Overall, the proposed framework offers various benefits to different stakeholders. Clients can realize significant improvements in time, cost, and quality through enhanced renovation construction project management. Third-party logistics companies can simplify the management of construction materials logistics by utilizing the IT-based tracking and dispatch services provided by the platform. Furthermore, the use of RFID and wearable technology enables more efficient real-time monitoring and cross-border logistics execution, facilitating logistics companies in

providing better services to construction companies by offering accurate information and supporting decision-making.

## ACKNOWLEDGMENT

RGC GRF 17203620, Guangdong Special Support Talent Program—Innovation and Entrepreneurship Leading Team under Grant 2019BT02S593, RGC Research Impact Fund (R7036-22), RGC Theme-based Research Scheme (T32-707-22-N) and HK Teaching Development Grant (TDG) Award (Project No. 951).

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
