# OpenReview forum: "Digital twin-enabled supply chain management with visibility and traceability: a case study"
_IEEE.org/ICIST/2024/Conference — IEEE ICIST 2024 Conference Submission_

### Official Review · Reviewer_iDwM · 2024-08-21
**accept**

**Rating:** 7
**Confidence:** 3

**Review:**

This paper proposed a digital twin-supported supply chain management framework that enhances visibility and traceability. The theory is correct and can be accepted after responding the following comments.
(1)	In the introduction, it is not enough to state the current work. It should be expended and reconstructed.
(2)	There are many typos and grammar errors. The authors should have a native English speaker or software packages to perform the editing check.
(3)	The conclusion of the article suggests using the present perfect tense for description.

---

### Official Review · Reviewer_TQLr · 2024-08-22
**The topic of this paper is interesting. Below is a list of comments that should be taken into account further when revising the paper.**

**Rating:** 8
**Confidence:** 3

**Review:**

the article offers a valuable contribution to the field of supply chain management in construction through its innovative use of digital twin technology. However, the complexity and potential data management issues suggest that further research and practical refinement are needed to fully realize its potential. The topic of this paper is interesting. Below is a list of comments that should be taken into account further when revising the paper.
1.	How does your paper significantly go beyond existing results?
2.	A lot of literature is cited in the article, but are there any recent relevant studies that have not been cited and discussed?It is recommended to supplement the latest research results to ensure the comprehensiveness of the literature review. It is recommended to supplement the latest research results to ensure the comprehensiveness of the literature review.
3.	There are the following grammatical and spelling errors in the article, and the structure of some sentences is not clear.

---

### Official Review · Reviewer_qCnH · 2024-08-27
**This paper proposes a digital twin-supported supply chain management framework that enhances visibility and traceability.**

**Rating:** 7
**Confidence:** 3

**Review:**

a The abstract should mainly include elements such as research purpose, methods and final results, and reviewers suggest optimizing the content of the abstract.
b There are some grammatical mistakes and typos. Please examine the full text further and revise them.
c What are the significant differences between this study and previous studies? The author needs more explicit emphasis.

---

### Decision · Program_Chairs · 2024-09-06

Accept (Oral)